# Determinants for α4β2 vs. α3β4 Subtype Selectivity of Pyrrolidine-Based nAChRs Ligands: A Computational Perspective with Focus on Recent cryo-EM Receptor Structures

**DOI:** 10.3390/molecules26123603

**Published:** 2021-06-12

**Authors:** Francesco Bavo, Marco Pallavicini, Rebecca Appiani, Cristiano Bolchi

**Affiliations:** 1Dipartimento di Scienze Farmaceutiche, Universita’degli Studi di Milano, I-20133 Milano, Italy; francesco.bavo@sund.ku.dk (F.B.); marco.pallavicini@unimi.it (M.P.); rebecca.appiani@unimi.it (R.A.); 2Department of Drug Design and Pharmacology, University of Copenhagen, DK-2100 Copenhagen, Denmark

**Keywords:** (*S*)-**nicotine**, nAChR, N-methyl-pyrrolidinyl, selectivity, alpha4beta2, alpha3beta4, docking, benzodioxane, cryo-EM

## Abstract

The selectivity of α4β2 nAChR agonists over the α3β4 nicotinic receptor subtype, predominant in ganglia, primarily conditions their therapeutic range and it is still a complex and challenging issue for medicinal chemists and pharmacologists. Here, we investigate the determinants for such subtype selectivity in a series of more than forty α4β2 ligands we have previously reported, docking them into the structures of the two human subtypes, recently determined by cryo-electron microscopy. They are all pyrrolidine based analogues of the well-known α4β2 agonist *N*-methylprolinol pyridyl ether A-84543 and differ in the flexibility and pattern substitution of their aromatic portion. Indeed, the direct or water mediated interaction with hydrophilic residues of the relatively narrower β2 minus side through the elements decorating the aromatic ring and the stabilization of the latter by facing to the not conserved β2-Phe119 result as key distinctive features for the α4β2 affinity. Consistently, these compounds show, despite the structural similarity, very different α4β2 vs. α3β4 selectivities, from modest to very high, which relate to rigidity/extensibility degree of the portion containing the aromatic ring and to substitutions at the latter. Furthermore, the structural rationalization of the rat vs. human differences of α4β2 vs. α3β4 selectivity ratios is here proposed.

## 1. Introduction

The development of ligands able to enhance the function of brain nicotinic acetylcholine receptors (nAChRs) has been, and still is, a pursued strategy to approach the treatment of cognitive deficits resulting from neurological and psychiatric disorders and some drug dependences [1]. The α4β2 nAChR, the major subtype in the brain, is primarily involved in such a strategy as its ligands, full or partial agonists, have shown a therapeutic potential as, for instance, antidepressants and drug cessation aids [2,3,4]. However, structural features relevant for full or partial agonism and for selectivity over the ganglionic α3β4 nicotinic subtype, which both condition the therapeutic range, remain two topical issues in SAR analysis and design of α4β2 ligands [5,6,7], undoubtedly more challenging than α4β2-α7 selectivity, which is generally more inherent in both highly affinitive α4β2 and α7 ligands [6,8]. On this matter, the structures of both human α4β2 and α3β4 receptors, recently determined by cryo-electron microscopy, can be a potent investigation tool suggesting new principles of agonist efficacy and, even more, of subtype selectivity [9,10].

The α3β4 subtype is mainly localized in the peripheral nervous system [11], where it mediates the effects on temperature, locomotor activity, and seizures elicitation and is also believed to be partially responsible for the cardiovascular and gastrointestinal liabilities of nicotine [12,13]. This subtype is highly concentrated also in a few brain regions and recent studies have suggested its role in influencing some behavioural effects of nicotine [14,15].

Structurally, nAChRs are transmembrane pentameric ligand-gated ion channels and are composed by five subunits assembled over a central aqueous pore, permeable to cations. Of particular note, until the recent resolution of the first full-length X-ray crystal structure the α4β2 nAChRs, three-dimensional information about the transmembrane domain, the pore structure and the binding site could only been obtained through homology modelling approaches [16,17,18,19]. Although models of the extracellular domain of nAChRs were considered reliable since based on X-ray crystal structures of the similar and homologous soluble acetylcholine binding protein (AChBP), the full length X-ray and cryo-EM structures of the human α4β2 and α3β4 nAChR substantially surpassed them. 

In both subtypes, as observed in the cryo-EM structures and illustrated in Figure 1A, the orthosteric ligand ((*S*)-**nicotine**) binds in the extracellular domain (ECD) at each α-β interface (AB and DE interfaces). Each subunit contributes to enclosing the ligand with three loops: Loops A-C from the α subunit (principal side) and loops D-F from the β subunit (complementary side). The pyrrolidine ring of (*S*)-**nicotine** is buried in the so-called “aromatic box”, a lipophilic cavity paved by loop A (α4-Tyr100 or α3-Tyr93) and D (β2-Trp57 or β4-Trp59) and surrounded by loop B (α4-Trp156 or α3-Trp149) and loop E (β2-Leu121 or β4-Leu123), which defines the back walls. The binding site is frontally closed by the flexible loop C (α4-Tyr197 and α4-Tyr204 or α3-Tyr190 and α3-Tyr197). Loop C packs on (*S*)-**nicotine** more tightly in the α3β4 subtype, rather than in the α4β2, making the latter more compact. The top wall is lined by loop E (β2-Val111 and β2-Phe119 or β4-Ile113 and β4-Leu121). In addition to interacting through lipophilic contacts, the positively charged pyrrolidine establishes π-cation interactions with the aromatic box and a charge-assisted H-bond with the backbone carbonyl of α4-Trp156 or α3-Trp149. A structural water molecule has been experimentally demonstrated to bridge through H-bonds the pyridine nitrogen of (*S*)-**nicotine** with the backbone NH of Leu121, and was observed in the cryo-EM 6PV7 and in the previously reported X-ray crystal structure of an acetylcholine binding protein/nicotine complex [20,21].

In the last 15 years, we have designed and developed some series of chiral α4β2 ligands, full and partial agonists and antagonists [22], initially linking the N-methyl-2-pyrrolidinyl residue, typical of nicotinoids, to C(2) of 1,4-benzodioxane [23,24], a scaffold widely employed to design bioactive molecules [25,26,27,28,29] and, in this instance, to mimick the aryloxymethyl portion of prolinol aryl ethers, well known high-affinity α4β2 ligands such as **A**-**84543** [30] (Figure 1B; for benzodioxane scaffold numbering see (*S*,*R*)-**21** formula). Successive steps were the decoration of the benzodioxane by introducing substituents at its C(7) [31], deconstruction of the dioxane ring to give new phenyl and pyridyl ethers of prolinol [7,32], replacement of benzene with pyridine to give the four regioisomeric pyridodioxanes [5], and again benzodioxane decoration with substituents at C(6) and C(5) [33]. In each of these series of prolinol aryl ethers or benzodioxane/pyridodioxane derivatives some compounds, as exemplified in Figure 1, exhibited one to hundred nanomolar α4β2 affinity and some of these also from good to high functional and binding selectivity over the α3β4 subtype. Such diversified results for a large number of chiral compounds with relatively similar structures prompted us to study their interactions at the binding sites of the two human α4β2 and α3β4 receptors by docking them into the respective structures determined by cryo-electron microscopy (cryo-EM 6CNJ and 6PV7) [9,10]. 

Here, we report the results of this analysis, which can provide a rational support to the discussion of the α4β2/α3β4 selectivity structural requirements, a still open and debated issue in the nicotinic ligands research.

## 2. Results and Discussion

### 2.1. Nicotine, Pyridyl Ethers of (S)-N-Me-Prolinol, and Inter-Species α4β2 vs. α3β4 Selectivity Ratios

#### 2.1.1. Analysis of α4β2 and α3β4 Affinity, Activity, and Selectivity Data from the Literature

(*S*)-**nicotine** (Figure 2), the most known pyrrolidine-based ligand of the nAChRs, is a competitive full agonist at all the nicotinic subtypes, with preference at the heteromeric α2-6- and β2-4-containing subtypes rather than to homomeric α7 and heteromeric α9α10 receptors. Particularly, (*S*)-**nicotine** is a nanomolar binder at the α4β2 subtype (K_i_ = 2 nM at human α4β2 [34], K_i_ = 4 nM at rat brain homogenates [32], and K_i_ = 10 nM at rat α4β2 [35]), while it only has submicromolar affinity at the α3β4 subtype (K_i_ = 261 nM at human α3β4 [32] and K_i_ = 440 nM at rat α3β4 [35]), with an approximate α4β2 vs. α3β4 selectivity ratio (defined as K_i_(α3β4)/K_i_(α4β2)) of 65 times [32]. These values have been determined in [^3^H]-epibatidine competition binding experiments and are in the same range of others reported in the literature [36,37].

At the beginning of the 1990s, the company Abbot developed and reported many pyrrolidine- and azetidine-based analogs of (*S*)-**nicotine**, where the pyridine ring was eitherreplaced by other aromatic rings, such as in **ABT**-**418** (bioisosteric 3-methyl isoxazole, Figure 2) [38] or distanced from the alicyclic core by a methylenoxy linker. The latter approach, reported by Abreo et al. in 1996, provided a series of 3-pyridyl ethers with nano- or subnanomolar affinity for the rat α4β2 subtype in the [^3^H]-(-)-cytisine displacement assay, among which compounds **A**-**84543** and **A**-**85380** (Figure 2) stood out as potent α4β2 full agonists with moderate functional selectivity over the α3β4 subtype [30]. The binding affinities at the rat α4β2 and α3β4 nAChRs in [^3^H]-epibatidine competition binding experiments were later measured for **A**-**84543** (K_i_ = 1.9 and 1400 nM, respectively) and for some novel 5-substituted analogs, among which (*S*)-**1** (K_i_ = 0.85 and 63,000 nM, respectively, Figure 2) [39]. The resulting very high to extraordinary α4β2 vs. α3β4 selectivity ratios (737 times for **A**-**84543** and 74,118 times for (*S*)-**1**) were however referred to rat nAChRs. Inter-species variations of α4β2 vs. α3β4 selectivity were later investigated by evaluating the α4β2 and α3β4 binding affinity and selectivity ratios of **sazetidine**-**A** (Figure 2), an optimized derivative of (*S*)-**1**. Although the selectivity trends were respected, **sazetidine**-**A** had a much higher α4β2 vs. α3β4 selectivity ratio at the rat subtype (24,000 times) than at the human (208 times) [40,41]. Interestingly, the difference could be ascribed to a much higher binding affinity of **sazetidine**-**A** at the human α3β4 nAChR (K_i_ = 52 nM) compared to the rat α3β4 nAChR (K_i_ = 10,000 nM). Instead, the binding affinities of **sazetidine**-**A** at the human α4β2 nAChR (K_i_= 0.64 nM) and at the rat α4β2 nAChR (K_i_ = 0.41 nM) were almost identical. 

In 2007, Young et al. identified by mutational studies two α3β4 amino acids (rβ4-Ser57 and rβ4-Ile58, corresponding to hβ4-Asn57 and hβ4-Val58, according to Uniprot numbering) conferring species-selectivity functional activity of **TMAQ** (Figure 2), a human α3β4 nAChR agonist with null activity at the rat isoform [42]. Competition binding experiments at hα3rβ4 and hα3hβ4 also demonstrated that these amino acids were responsible for a modest but significant (14-fold) difference in the affinity of binding of **TMAQ** to nAChRs containing the human and rat α4 subunit. Similarly, in 2015, Tuan et al. reported **AT**-**1001** (Figure 2) as a human selective α3β4 nAChR ligand with similar K_i_s at human and rat α4β2 nAChRs, while 20 times higher affinity at the human than rat α3β4 nAChRs [43].

#### 2.1.2. Structural Rationalization of the Rat vs. Human Differences of α4β2 vs. α3β4 Selectivity Ratios

The much higher difference in affinity between the human α3β4 and the rat α3β4 of **sazetidine**-**A** compared to that of **TMAQ** and **AT**-**1001**, (192-times vs. 14- and 20-times), also reflected in the 115 times higher α4β2 vs. α3β4 selectivity at the rat compared to the human subtypes, prompted us to hypothesize that other structural inter-species differences of the α3β4 subtype could be involved. Differently, the similar binding affinities at the human and rat α4β2 subtypes of **sazetidine**-**A** and of our *in-house* ligands, among which (*S*,*R*)-**2**, suggest a very high degree of inter-species binding site similarity of the α4β2 subtype [33].

To address this question, we inspected the recently reported cryo-EM of (A) the human α4β2 nAChR complexed with (*S*)-**nicotine** (6CNJ) and of (B) the human α3β4 nAChR complexed with (*S*)-**nicotine** (6PV7), and we analyzed it together with the human/rat α4, β2, α3, and β4 alignments (Figure 3). (A) At the α4β2 nAChR, no amino acid differences were observed between rat and human receptors within 5 Å from (*S*)-**nicotine**, explaining why the K_i_s measured at the rat and human α4β2 are almost identical and can be considered as surrogates. (B) Instead, within 5 Å from (*S*)-**nicotine** at the human α3β4 subtype, the residue hβ4-Leu121 and hβ4-Leu112 are not conserved in the rat species, where they are replaced by rβ4-Gln121 and rβ4-Val112, respectively. Since the side chain of the latter (rβ4-Val112) is not pointing at the binding site (not shown), it was not further considered. Interestingly, the residue corresponding to hβ4-Leu121 is a Phe119 at the hβ2 subtype and has been proposed to stabilize (*S*)-**nicotine** in the hα4β2 binding site by π-π interactions with its pyridine ring. Based on these observations, we hypothesize that the residues rβ4-Gln121, hβ4-Leu121, and hβ2-Phe119 are strongly affecting binding affinities (and consequently, selectivity ratios) at rα3β4, hα3β4, and hα4β2 nAChRs, respectively. 

To support our hypothesis, we performed molecular docking of the pyridyl ether **A**-**84543** at the hα4β2 and hα3β4 binding sites, extracted, aligned, and refined, respectively from the cryo-EMs 6CNJ and 6PV7. In particular, the structural water molecule known to be critical for (*S*)-**nicotine** activity at the hα4β2 binding site and not detected in the cryo-EM, was extracted from hα3β4 and included in the hα4β2 binding site [20]. Additionally, we also docked **A**-**84543** into a model of the rα3β4 orthosteric binding site prepared from the hα3β4 binding site of 6PV7 by the in silico site directed mutagenesis of hβ4-Leu121 into rβ4-Gln121 (Figure 4A). In both human subtypes, **A**-**84543** positions the positively charged N-methyl pyrrolidine ring within the so-called aromatic box (Tyr197, Tyr204, Trp156, and Trp57 at the α4β2 nAChR; Tyr190, Tyr197, Trp149, and Trp59 at the α3β4 nAChR), with a suitable orientation of the N-methyl-pyrrolidinyl ring for establishing a charge assisted H-bond with the backbone carbonyl of Trp (hα4-Trp156 and hα3-Trp149). Both pyridine nitrogens interact as HBAs with the structural water molecule. However, we observed a 40° plane drift between the two aromatic rings associated with a RMSD of 1.1540 Å, plausibly since the hα4β2-Phe119 stabilizes the ligand through face-to-edge π-π interactions, while the hα3β4-Leu121 can only contribute with VdW contacts. The lack of a stabilizing π-π interaction causes the pyridine ring to “fall out” from the plane where the corresponding ring of (*S*)-**nicotine** is placed (the corresponding angle is only of 14°, with a RMSD of 0.6082 Å when (*S*)-**nicotine** from the two original cryo-EMs are compared, Figure 3A). Additionally, the hα3β4 binding site is less compact due to a 2.1 Å outward displacement of loop C, which may also contribute to a lower binding affinity at the hα3β4 and to the generally good hα4β2 vs. hα3β4 selectivity of pyridyl ether nicotinic ligands. Interestingly, when **A**-**84543** is docked at the rα3β4, where hβ4-Leu121 is replaced by the more flexible and hydrophilic rβ4-Gln121, it bends with an even wider angle of 80° reaching an overall RMSD of 2.5002 Å, disallowing the conventional interactions with the binding site. These observations, taken together, could explain why pyridyl ether-based nicotinic ligands have a much higher α4β2 vs. rα3β4 selectivity than hα4β2 vs. hα3β4 selectivity ratios.

Since the cryo-EM of the complex **AT**-**1001**/hα3β4 is also available (6PV8), and **AT**-**1001** has been experimentally proven to bind differently at the two apparently identical hα3β4 interfaces (AB and DE interfaces), we also prepared the rα3β4 models based on both interfaces by in silico mutagenesis and docked **AT**-**1001**. The binding poses of **AT**-**1001** at the rα3β4 binding site based on the AB interface is nearly identical to that at the hα3β4 from the cryo-EM (RMSD of 0.5936 Å). When docked at the rα3β4 binding site based on the DE interface, the aromatic ring of **AT**-**1001** bends inwards of around 30° (RMSD of 0.9412 Å). The less dramatic change in **AT**-**1001** positioning, occurring at one binding interface only, could account for the lower inter-species ratios in α3β4 binding affinity (Figure 4B,C). An overview of the relevant interactions is reported in Appendix A.

### 2.2. Rationalization of Determinants for α4β2 vs. α3β4 nAChR Selectivity of Small, Flexible Phenyl Ethers of (S)-N-Methyl-Prolinol

Between 2015 and 2016, Bolchi et al. reported a series of unichiral *m*-substituted phenyl ethers of (*S*)-N-methyl prolinol, compounds (*S*)-**3**–**7**, as flexible analogs of the rigid α4β2 partial agonist (*S*,*R*)-**2**, that binds the α4β2 nAChR with nanomolar affinity (K_i_ = 0.012 μM) and has a moderate rα4β2 vs. hα3β4 selectivity ratio of 26 times (Table 1) [7,32]. Among these, (*S*)-**3** can also be considered as the hydroxylated derivative of compound (*S*)-**4**, reported by Elliot in 1997 as a binder of rα4β2 nAChR (K_i_ = 0.042 μM) [44]. 

Deconstruction of the dioxane ring of (*S*,*R*)-**2** into (*S*)-**3** switched the functional activity from hα4β2 partial agonist to full agonist, and increased both binding affinity (K_i_ = 0.0011 μM) and the rα4β2 vs. hα3β4 selectivity ratio (67-fold). Methylation of the hydroxyl group to give (*S*)-**5** caused a sharp drop in α4β2 affinity (K_i_ = 0.600 μM, 400 times), while replacement with a nitro group in (*S*)-**6** decreased affinity (K_i_ = 0.0312 μM, 28 times).

To understand the structural determinants underlying the trends in α4β2 affinity and α4β2 vs. α3β4 selectivity, we docked compounds (*S*)-**3**, (*S*)-**4**, (*S*)-**5,** and (*S*)-**6** in the α4β2 and in the α3β4 binding sites, extracted and refined, respectively from the cryo-EM structures 6CNJ and 6PV7. In detail, since none of the compounds contained a pyridine nitrogen as a suitable hydrogen bond partner for the structural water molecule, and all except (*S*)-**4** presented a substituent in the same (*meta*) position instead, we removed the water molecule from the α3β4 binding pocket. This approach was previously successful in describing the binding mode of (*S*,*R*)-**2** at α4β2 and α3β4 nAChRs [33]. 

(*S*)-**3** binds similarly at the interfaces of both subtypes and coherently overlays the pyrrolidine ring with that of (*S*)-**nicotine** and (*S*,*R*)-**2** (Figure 5A). Furthermore, the *m*-hydroxyl group of (*S*)-**3** superimposes with the structural water molecule, and interacts with the backbone carbonyl of β2-Asn109/β4-Asn111 and the backbone NH of β2-Leu121/β4-Leu123 within the small and hydrophilic pocket known to be critical for nAChR affinity [33]. The 3-times higher affinity of (*S*)-**3** than (*S*)-**nicotine** might be attributed to the replacement of a flexible water-mediated H-bond network with the more rigid and stable direct interaction between the *m*-OH and the receptor. (*S*)-**3** shares the same α4β2 vs. α3β4 selectivity ratio than (*S*)-**nicotine**, and similarly to the latter, the phenolic ring of (*S*)-**3** is 20° bent outwards due to the hβ2-Phe119 to hβ4-Leu121 replacement mentioned earlier (RMSD of 1.0536 Å).

When the hydroxyl group of (*S*)-**3** is replaced by a bulkier nitro group in (*S*)-**6**, the ligand is forced to assume a kinked conformation, but still interacts, although incompletely, with the backbone of β2-Leu121. Instead, the less polar and bulkier methoxy group of (*S*)-**5** sterically clashes with β2-Leu121, while the naked (*S*)-**4** lacks any feature able to reach and interact with the small hydrophilic subpocket (Figure 5B). An overview of the relevant interactions is reported in Appendix A.

### 2.3. Binding Modes of the α4β2 Superagonist Hydroxy Pyridyl Ether of N-Methyl Prolinol (S)-7

(*S*)-**7** combines in its structure the *meta*-hydroxylation of (*S*)-**3** and the piridyl ring of **A**-**84543**, both crucial to reach and interact with β2-Asn109 and β2-Leu121, either directly or by bridging a water molecule [20]. As a result, the hydroxyl piridyl ether (*S*)-**7** has nanomolar α4β2 binding affinity (K_i_ = 0.0037 μM) and a α4β2 vs. α3β4 selectivity ratio similar to that of (*S*)-**3** (63 times) (Table 1). Whereas (*S*)-**3** is a full agonist with one digit micromolar potency, (*S*)-**7** is functionally an α4β2 superagonist, with high efficacy (661% than the efficacy of Ach 1 mM) but low potency (EC_50_ = 73 μM). Instead, it has low efficacy at the α3β4 nAChR (15% than the efficacy of Ach 1 mM), and therefore is a functionally selective α4β2 superagonist. 

Although it is difficult to describe the functional activity at nAChRs by computational techniques only, we docked (*S*)-**7** at the α4β2 and α3β4 binding sites to propose a plausible binding mode and to suggest hypotheses on superagonism and functional α4β2 vs. α3β4 selectivity of (*S*)-**7**. 

Due to its structural features, (*S*)-**7** can hypothetically (A) establish hydrogen bonds with the structural water molecule through the pyridine ring or (B) replace the water molecule and directly interact through the *meta*-hydroxyl or (C) both. Therefore, we docked (*S*)-**7** at both binding pockets, both including and removing the structural water. As expected, (*S*)-**7** assumes two distinct binding poses at the α4β2 nAChR, depending on the presence or absence of water (Figure 6A). When the structural water is included, (*S*)-**7** folds onto itself and the pyridine nitrogen interacts with water, strongly resembling the binding mode of **A**-**84543**. Instead, when the structural water is not included, (*S*)-**7** binds with an extended conformation, similar to that of (*S*)-**3**, and places the hydroxyl group within the small hydrophilic pocket. Of particular note, when (*S*)-**7** binds with a folded and **A**-**84543**-like conformation, the *meta*-OH establishes an hydrogen bond with α4-Tyr204. Instead, when (*S*)-**7** binds with an extended (S)-4-like conformation, the pyridine nitrogen is placed nearby α4-Thr157, whose hydroxylated side chain is stabilized backwards by H-bond with β2-Asn109. Under the assumption of receptor flexibility, we scanned the rotamers of α4-Thr157 and observed that the third most populated rotamer according to the Schrodinger rotamer library exposed the hydroxyl group at less than 4 Å from the nitrogen, suggesting the potential for hydrogen bond.

Although it is tempting to speculate that either the dual binding mode of (*S*)-**7** or the additional hydrogen bonds with α4-Tyr204 or α4-Thr157 could be correlated with the superagonist activity, mutational studies would be needed to further investigate these hypotheses. At the water-free α3β4 nAChR, (*S*)-**7** binds with an extended conformation similar to (*S*)-**3** (Figure 6B). At the α3β4 binding site, it still folds but looses contact with the β subunit, interacting with α3-Ser150 instead. The similar binding pose of (*S*)-**7** and (*S*)-**3** at the water-free binding sites could explain the same α4β2 vs. α3β4 selectivity ratio (RMSD of 1.6933 Å), while the disruption of a critical interaction between the water-free α3β4 binding site and the folded conformation of (*S*)-**7** could be correlated to the very low efficacy at this subtype, which makes (*S*)-**7** functionally selective. An overview of the relevant interactions is reported in Appendix A.

### 2.4. Semirigid Pyrrolidine-Based Nicotinic Ligands: Rationalization of Determinants for α4β2 nAChRs Affinity and for High α4β2 vs. α3β4 Selectivity of (S,R)-14

In the same work, we also reported a series of semi-rigid analogs of (*S*)-**3**, (*S*)-**4**, and **A**-**84543**, namely **8**–**14**, where either a methyl or a methoxy group was introduced to impose conformational strains, reduce flexibility, and mimic the rigid core of the partial agonist (*S*,*R*)-**2** (Table 2). Both analogs of (*S*)-**4**, ((*S*)-**8** and (*S*)-**10**), had micromolar affinity at the α4β2 nAChR. The α4β2 affinity of compounds derived from (*S*)-**3**, ranged from 2-digits nanomolar (K_i_ = 18.9 and 11.1 nM, for (*S*)-**9** and (*S*,*R*)-**11**, respectively) to low submicromolar (K_i_ = 0.192 μM for (*S*,*S*)-**11**). Among the analogs of **A**-**84543**, only (*S*,*R*)-**14** had nanomolar α4β2 affinity (K_i_ = 27 nM).

All the semi-rigid compounds were docked at the α4β2 binding sites optimized from 6CNJ for rigid benzodioxane-based ligands as reported in Bavo et al. [33]. Lacking any feature required either displacing or interacting with the structural water molecule, (*S*)-**8**, (*S*,*R*)-**10**, and (*S*,*S*)-**10** cannot establish any productive interaction with the β2 subunit (Figure 7A). Instead, (*S*)-**9**, (*S*,*R*)-**11**, and (*S*,*S*)-**11**, reflecting their nanomolar α4β2 K_i_s, directly bind the β2 subunit with the *m*-OH substituent (Figure 7B). Of these, (*S*,*S*)-**11** shows some unfavorable steric contacts between the methyl group and the side chains of β2-Leu121 and α4-Cys199, whereas the methyl substituent in (*S*,*R*)-**11** and the methoxy group of (*S*)-**9** are accommodated in a more spacious area of the binding site. Among the semi-rigid analogs of **A**-**84543**, we observed a productive orientation of the pyridyl ring only in the submicromolar binders (*S*)-**13**, (*S*,*R*)-**14**, and (*S*,*S*)-**14**, while (*S*)-**12**, with micromolar K_i_, does not interact with the structural water molecule (Figure 7C). Again, whereas the methoxy and methyl substituents of (*S*)-**13** and (*S*,*R*)-**14** are accommodated in a quite spacious area, the methyl group of the weaker binder (*S*,*S*)-**14** clashes with the side chain of α4-Cys199. Of outmost importance, (*S*,*R*)-**14** was the most α4β2 selective compound of the series, with an α4β2 vs. α3β4 selectivity ratio of 185 times, while the less affinitive epimer (*S*,*S*)-**14** was also not selective (6 times only). Hence, (*S*,*R*)-**14** was also docked at the α3β4 binding site optimized from 6PV7 as reported in Bavo et al. (Figure 7D) [23]. Whereas (*S*,*R*)-**14** is stabilized at the α4β2 nAChR by π-π stacking with β2-Phe119, at the α3β4 the pyridyl ring bends out with a wide angle of 51° and an overall RMSD of 2.7792 Å, loosing the interaction with the structural water molecule. The same effect was evident for (*S*,*R*)-**33** (discussed later), its fully rigidified analogue previously reported [5,33]. Furthermore, the binding conformation of (*S*,*R*)-**14** at the α4β2 nAChR superimposes well with that of (*S*,*R*)-**33**, positioning the α-methyl functional group in the area where the corresponding C3 of the benzodioxane is placed (compare Figure 7D with Figure 10B). Likewise, at the α3β4 nAChR, the two ligands share a predicted similar conformation and binding pose, where the critical interaction between the pyridyl nitrogen and the structural water is lost, and in which the α-methyl group of (*S*,*R*)-**14** is positioned very close to α3-Tyr197, impairing α3β4 binding. An overview of the relevant interactions is reported in Appendix A.

### 2.5. Structural Determinants for α4β2 nAChR Affinity and α4β2 vs. α3β4 Selectivity of 5-Substituted 3-Hydrophenyl Ethers of N-Methyl Prolinol

In 2016, we developed a series of 5-substituted derivatives of (*S*)-**3** and of its analog (*S*)-**6** ((*S*)-**15**–**20**), aiming to more α4β2 vs. α3β4 selective compounds (Table 3) [7]. Coherently with the literature, compounds (*S*)-**15** and (*S*)-**18**, bearing a 6-hydroxy-1-hexynyl substituent at the 5-position, were two times more selective than (*S*)-**3** and (*S*)-**6,** respectively. Unexpectedly, the slightly shorter but still hydroxylated *p*-hydroxyphenyl moiety of (*S*)-**16** and (*S*)-**19** increased the binding affinity at the α3β4 nAChR than the parent compounds (*S*)-**3** (2.4 times) and (*S*)-**6** (7.7 times), causing an overall decrease of the α3β4/α4β2 selectivity ratios.

To interpret these data, we docked compounds (*S*)-**15**–**20** to the α4β2 and α3β4 binding sites, previously adapted to the shape and size of (*S*)-**16,** the compound with the highest α4β2 and α3β4 binding affinities of the series. The results revealed a diverse binding mode at the two subtypes. At the α4β2 nAChR, (*S*)-**15**–**16** and (*S*)-**18**–**19** are stabilized by the conventional interactions: the pyrrolidine ring is placed within the aromatic box, the *m*-substituent interacts with the β subunit, and the aromatic ring establishes π-π stacking with β2-Phe119. Consequently, the *p*-hydroxyphenyl and the hexynol substituents are oriented towards the relatively small and lipophilic β2-Val111, and carry the phenolic group and the hydroxyl group between α4-Glu202 and the non-conserved β2-Lys79 (Figure 8A). Removal of the phenolic function to compound (*S*)-**17** and (*S*)-**20** deprives the two charge assisted H-bonds, and accounts for the up to 140 times loss of α4β2 binding. 

The proposed binding modes of the non-selective (*S*)-**16** and of the selective (*S*)-**15** at the α3β4 subtype are instead different (Figure 8B). As previously hypothesized for (*S*)-**3**, the replacement of β2-Phe119 with β4-Leu121 causes the phenyl moieties to bend towards β4-Leu123. Additionally, the α4β2 residue β2-Val111 is replaced by the bulkier β4-Ile113, that prevents the 5-hydroxyphenyl and the 5-hexynol substituents to reach α3-Glu195 and β4-Ile81, corresponding to the pair α4-Glu202/β2-Lys79. As a result, the 5-*p*-hydroxyphenyl and the 5-hexynol substituents are accommodated between the disulphide bridge of α3-Cys193/Cys194 and β4-Leu121. The phenolic function of (*S*)-**16** interacts with the non-conserved α3-Glu194 and β4-Arg115 (corresponding to α4-Ala201 and β2-Ser113, respectively) and its removal disrupts the hydrogen bond network, causing a drop in α3β4 affinity. On the other hand, the 5-hexynol substituent of (*S*)-**15** stretches outside from the binding pocket and looses the hydrogen bond network with α3-Glu194 and β4-Arg115. An overview of the relevant interactions is reported in Appendix A. 

### 2.6. Structural Determinants for α4β2 nAChR Affinity and α4β2 vs. α3β4 Selectivity of 7-Substituted and Unsubstituted N-Methyl Pyrrolidinyl-Benzodioxanes 

In 2009, we reported the synthesis of all the stereoisomers of 2-(2-pyrrolidinyl)benzodioxanes (**21**) and evaluated their binding affinities at the α4β2 nAChRs (Table 4) [24]. The two stereoisomers with absolute configuration “R” at the pyrrolidine stereocenter were devoid of affinity, while (*S*,*S*)-**21** and (*S*,*R*)-**21** had submicromolar α4β2 K_i_s of 0.47 μM and 0.26 μM, respectively. Later, they were shown to be micromolar binders at the α3β4 [5]. In 2011, we developed the series of 7-substituted analogs **2**, **22**–**28** (Table 4), among which only (*S*,*R*)-**2** was able to bind at the α4β2 nAChR (K_i_ = 0.012 μM). 

Our proposed binding mode at the α4β2 receptor for (*S*,*R*)-**2** and for its amino analog (*S*,*R*)-**29** (reported later, together with **30**–**32**), easily explains the lack of affinity of any other 7-substituted analogs. The rigid and extended benzodioxane core correctly positions the 7-OH or 7-NH_2_ group within the small hydrophilic binding pocket lined by β2-Asn109 and β2-Leu121, while any other bulkier group cannot be accommodated (Figure 9A). At the slightly larger binding site of the α3β4 nAChR, the more rigid, more extended, and better stabilized (*S*,*R*)-**2** (RMSD of 1.3140 Å) and (*S*,*R*)-**29** (RMSD of 1.1703 Å), are less influenced by the Phe to Leu replacement than the flexible (S)-**3**, resulting only in a moderate to null α4β2 vs. α3β4 selectivity (Figure 9B). To address the recurring α4β2 stereopreference for the (*S*,*R*) configuration, typical for both the unsubstituted and the 7-substituted analogs, we performed a conformational search of (*S*,*R*)-**2** and (*S*,*S*)-**2**, and rigidly docked the resulting set of conformers at the α4β2 binding site, prepared as previously described. Although they both bind, (*S*,*S*)-**2** binds in a high energy conformation (relative potential energy of 10.070 kJ/mol), while (*S*,*R*)-**2** binds in the most energetically favored conformation, likely explaining the 35 times difference in binding affinity (not shown). An overview of the relevant interactions is reported in Appendix A.

### 2.7. Structural Determinants for α4β2 nAChR Affinity and α4β2 vs. α3β4 Selectivity of N-Methylpyrrolidinyl Pyridodioxanes

In 2017, we developed **33**–**36**, the four regioisomeric pyridodioxane-based analogs of (*S*,*R*)-**21**, in both the (*S*,*R*) and the (*S*,*S*) diasteromeric forms (Table 5) [5]. (*S*,*R*)-**33** was the only compound retaining the submicromolar binding affinity at the α4β2 subtype (Ki = 0.41 μM), with a 40-fold α4β2 vs. α3β4 selectivity ratio. 

Relying on the proposed binding mode of (*S*,*R*)-**33**, we docked (*S*,*R*)-**33**–**36** at the α4β2 and α3β4 binding sites optimized as precedently described [33]. Reflecting the binding data, only (*S*,*R*)-**33** consistently orients the N-Methyl-pyrrolidine ring within the aromatic box and interacts with the structural water (Figure 10A). The low affinity of (*S*,*R*)-**33** at α3β4 can be attributed to the replacement of β2-Phe119 with β4-Leu121, that causes the loss of a stabilizing π-π interaction and the ligand to bend in the binding site (ring angle of 60.3° and RMSD of 2.1924 Å). Although this difference is observed also in the flexible ligands (described before) and in (*S*,*R*)-**2**, it is much more evident for the rigid (*S*,*R*)-**33** and its closely related semirigid analogue (*S*,*R*)-**14** (Figure 10B, to be compared with Figure 7D). Indeed, the rigidity of the pyridodioxane core of (*S*,*R*)-**33** or the conformational strain imposed by the α-methyl group of (*S*,*R*)-**14**, combined with the H-bond interaction with the structural water causes the pyridodioxane and pyridyl moiety to face more tightly the β2-Phe119. An overview of the relevant interactions is reported in Appendix A.

### 2.8. Structural Determinants for α4β2 nAChR Affinity and α4β2 vs. α3β4 Selectivity of 5- and 6-Substituted N-Methyl Pyrrolidinyl-Benzodioxanes

To complete the investigation around the benzodioxane scaffold of (*S*,*R*)-**21**, we recently developed the 5-substituted analogs **37**–**40** and the 6-substituted analogues **40**–**44** (Table 6) [33].

Interestingly, (*S*,*S*)-**37** showed high nanomolar affinity at the α4β2 nAChR and a very high α4β2 vs. α3β4 selectivity ratio (100 times). According to the proposed binding mode, (*S*,*S*)-**37** twists its benzodioxane core to fit the small hydrophilic pocket with the amino substituent, in a similar fashion to (*S*,*R*)-**33** (Figure 11A). The larger nature of the α3β4 binding site and the β2-Phe119 to β4-Leu121 replacement are responsible for the high α4β2 vs. α3β4 selectivity ratio of (*S*,*S*)-**37** (ring angle of 232° and RMSD of 3.1030 Å). Interestingly, the stereopreference shown for (*S*,*S*)-**37** (discussed previously, [33]) is not conserved among the series. Aiming at rationalizing the α4β2 affinity/stereochemistry patterns, we docked **37**–**40** at the α4β2 binding site (Figure 11B). (*S*,*S*)-**38**, but not its diastereoisomer (*S*,*R*)-**38**, retraces the binding mode of (*S*,*S*)-**37**. However, the bulky nitro group cannot fully access β2-Asn109 and β2-Leu121, justifying the 36-fold difference in α4β2 affinity. The two diasteromers of **39**, which have a similar affinity, bind very similarly, while only the diasteromer (*S*,*R*)-**40** correctly places the pyrrolidine ring within the aromatic box.

Although none of the 6-substituted analogs reached submicromolar α4β2 affinity, we found their slight preference towards the α3β4 subtype quite curious. Therefore, we docked them at both binding sites, and analyzed the best scored binding poses (Figure 11C,D). Apparently, the benzodioxane core of (*S*,*R*)-**43** is oriented similarly to that of (*S*,*R*)-**2**, but the 6-substituent cannot access the subpocket and stands at the entrance instead. Therefore, in the rather small and narrow α4β2 binding site, the ligand is pushed backwards and induces a twist of the pyrrolidine ring. Such effect is not observed when (*S*,*R*)-**43** is docked at the α3β4 binding site: The room generated by the Phe to Leu replacement and the generally more spacious cavity allow the ligand to fit in a more relaxed conformation, without any backward drift, and to place the 6-substituent in an uninfluential area of the receptor. An overview of the relevant interactions is reported in Appendix A.

## 3. Materials and Methods

### 3.1. Ligand Preparation

All ligands were built with the 2D editor sketch in Maestro (Schrödinger LLC: New York, NY, USA, 2019) without specifying the absolute configuration of the stereocenters [45]. Ligprep was used with default settings to assign the correct ionization state and to generate all the possible stereoisomers [46]. Of note, upon protonation of the pyrrolidine nitrogen to an asymmetric ammonium, two different absolute configurations are possible. Only ligands with “R” configuration at the nitrogen were considered, as it is the same absolute configuration of (*S*)-**nicotine** at the cryo-EM 6CNJ and 6PV7. Stereochemistry of all the sterocenters was visualized and the stereoisomers for which no pharmacological data were available were discarded.

### 3.2. Binding Site Preparation: Water-Containing and Water-Free hα4β2 and hα3β4 from cryo-EM Structures

The hα4β2 and hα3β4 binding sites were extracted from the appropriate cryo-EM structure (6CNJ for hα4β2 and either 6PV7 or 6PV8 hα3β4). In detail, the pair of subunits enclosing the ligand ((*S*)-**nicotine** in 6CNJ and 6PV7, **AT**-**1001** in 6PV8) was extracted, truncated to preserve the extracellular domain only, and aligned. Since in 6PV8 the two interfaces (AB and DE) bind **AT**-**1001** with different poses, they were considered separately. Water-containing binding sites were obtained as follows: The structural water molecule bridging the pyridine nitrogen of (*S*)-**nicotine** and the β4 subunit, included in the hα3β4 of 6PV7, was copied and merged with the hα4β2 binding site. The resulting dimeric water-containing complexes were preprocessed, the hydrogen bond network optimized and subjected to constrained minimization using the Protein Preparation Wizard with default settings [47,48]. The water-containing binding sites prepared from 6CNJ and 6PV7 are hereby named for the sake of clarity 6cnj-wc-hα4β2 and 6pv7-wc-hα3β4, respectively. Removal of the structural water molecule from the water-containing binding site provided the corresponding water-free binding sites, named 6cnj-wf-hα4β2 and 6pv7-wf-hα3β4. The water-free binding sites originated from the two different interfaces of 6PV8 are named 6pv8-AB-hα3β4 and 6pv8-DE-hα3β4.

### 3.3. Binding Site Preparation: Water-Containing and Water-Free hα4β2 and hα3β4 from IFD Complexes

To account for the reduced flexibility and increased size of ligands **2**, **8**–**14**, **21**–**44**, the hα4β2 and hα3β4 binding sites reported in Bavo et al., were employed [33]. Briefly, the most rigid and bulky compounds with the highest binding affinity at the hα4β2 and hα3β4 nAChRs, namely (*S*,*R*)-**2** and (*S*,*R*)-**29**, were docked at the 6cnj-wf-hα4β2 and 6pv7-wf-hα3β4, using the Induced Fit Docking Protocol (default settings, except for scaling factor 1.0 and XP precision in the re-docking step) [49]. The resulting water-free binding sites are named 2ifd-wf-hα4β2 and 29ifd-wf-hα3β4, respectively. Removal of the ligand and inclusion of the structural water molecule, extracted from 6PV7, provided the water-containing binding sites named 2ifd-wc-hα4β2 and 29ifd-wc-hα3β4, respectively.

The same procedure was applied to adapt the hα4β2 and hα3β4 binding sites to the elongated shape of compounds (*S*)-**15**–**20**, by docking the ligand with the highest α4β2 and α3β4 binding affinity of the series, (*S*)-**16**, using the Induced Fit Docking Protocol [49]. The best scored hα4β2/(*S*)-**16** and hα3β4/(*S*)-**16** complexes were used for the subsequent docking steps, and named 16ifd-wf-hα4β2 and 16ifd-wf-hα3β4. 

### 3.4. Human/Rat Sequence Alignments

The primary sequences of the human and rat α4, β2, α3, and β4 subunits were retrieved by Uniprot (hα4 P43681, rα4 P09483, hβ2 P17787, rβ2 P12390, hα3 P32297, rα3 P04757, hβ4 P30926, rβ4 P12392) and aligned in human-rat pairs with Clustal Omega, with default settings [50,51]. The alignments were manually inspected with Aliview [52] in the region of the binding site, defined as the area within 10 Å from the ligand (*S*)-**nicotine** in 6CNJ and 6PV7. Inter-species differences were identified within 5 Å from (*S*)-**nicotine**.

### 3.5. Binding Site Preparation: Rα3β4 by In Silico Site Directed Mutagenesis

The human β4-Leu121 residue of 6pv7-wc-hα3β4, 6pv8-AB-hα3β4, and 6pv8-DE-hα3β4 was mutated in silico into rβ4-Gln121 and the rotamer more superimposable with the original side chain was selected (second populated rotamer not showing any steric clash), providing the binding sites named 6pv7-wc-rα3β4, 6pv8-AB-rα3β4, and 6pv8-DE-rα3β4.

### 3.6. Molecular Docking

Docking was performed using the Glide XP Ligand Docking Protocol docking in Maestro, Schrodinger, with default settings (van der Waals radius scaling factor: 0.8, partial cutoff: 0.15, XP precision, OPLS3e forcefield, flexible sampling), with a grid centered on the present ligand [53]. Since all the compounds presented in this work are analogues of (*S*)-**nicotine** and share with it an identical pyrrolidinyl moiety, they are expected to position it similarly to (*S*)-**nicotine**. Therefore, the highest ranked pose according to the glide model score that placed the positively charged nitrogen in the aromatic box, not further than 1.5 Å from the positively charged nitrogen of (*S*)-**nicotine**, was selected.

#### 3.6.1. Docking of A-84543, (*S*)-**nicotine**, AT-1001, and Compounds **2**–**43**

**A**-**84543** and (*S*)-**nicotine** were docked in the 6pv7-wc-rα3β4 binding site, while **AT**-**1001** was docked in the 6pv8-AB-rα3β4 and 6pv8-DE-rα3β4 binding sites.

The grids generated from 6cnj-wc-hα4β2 and 6pv7-wc-hα3β4 were used for docking of **A**-**84543** and (*S*)-**7**, and those originating from 6cnj-wf-hα4β2 and 6pv7-wf-hα3β4 were used for docking of (*S*)-**3**–**7**. Compounds **2**, **8**–**11**, **21**–**32**, and **37**–**44** were docked at the 2ifd-wf-hα4β2 binding site, while compounds **12**–**14** and **33**–**36** were docked in the 2ifd-wc-hα4β2 binding site. Similarly, compounds (*S*,*R*)-**26**, (*S*,*R*)-**29**, (*S*,*R*)-**31**, (*S*,*S*)-**37**, and (*S*,*R*)-**43** were also docked at the 29ifd-wc-hα3β4, and compounds (*S*,*R*)-**14**, (*S*,*R*)-**33**, were docked at the 29ifd-wc-hα4β2 binding site. Compounds (*S*)-**15**–**20** were docked at the 16ifd-wf-hα4β2 and 16ifd-wf-hα3β4 binding sites. 

Additionally, a conformational search of compounds (*S*,*R*)-**2** and (*S*,*S*)-**2** all conformers was performed using the torsional sampling (Monte Carlo Multiple Minimum) method with default settings [54]. All the resulting conformers where docked at 2ifd-wf-hα4β2, with Glide XP as described above, but using rigid ligand sampling rather than flexible.

#### 3.6.2. Self-Docking Studies

To exclude the possibility of excessive deformation of the binding pocket during the binding site preparation and to validate the docking procedure, we re-docked the native ligands in each prepared binding site using the same docking protocol used for our ligands. We considered them acceptable since the RMSD (root-mean square deviation) between the re-docked ligand and the original cryo-EM ligand was lower than 2 Å (Appendix A) [55]. 

## 4. Conclusions

Here, we have highlighted the structural determinants for α4β2 vs. α3β4 subtype selectivity in a large series of flexible, semi-rigid, and rigid analogues of the α4β2 vs. α3β4 selective prolinol pyridyl ether **A**-**84543** by docking at the binding sites of the two human nicotinic subtypes, whose structures have been recently determined by cryo-electron microscopy. 

Our analysis was necessarily based on the comparison of rα4β2 with hα3β4 binding affinities. While the α4β2 affinities of the title compounds are similar in rat and in man [33], the α3β4 affinities are generally much lower in rat than in man resulting in much higher α4β2 vs. α3β4 selectivity when the terms of comparison is the rα3β4 affinity rather than the relatively higher hα3β4 affinity. In this regard, the pyridyl ether **Sazetidine**-**A** is a case in point. Consequently, whereas the use of human α4β2 structures for docking is acceptable in any case, docking cannot be regardless of the use of human or rat α3β4 structure. To explain such a different behaviour, we evaluated the role, for the α3β4 affinity of **A**-**84543**, of the not-inter-species conserved position 121 in the β4 side (rβ4-Gln121 →hβ4-Leu121). This position seems to be crucial for the α4β2 vs. α3β4 subtype selectivity since it corresponds, in the hβ2 side, to Phe119, which is advantageously involved in a π-πstabilization with the pyridine ring. Well, we observed that the polar and flexible amino acid rβ4-Gln121 disfavoured the interaction of the **A**-**84543** pyridine ring with the β4 side even more than the rigid and lipophilic hβ4-Leu121, due to the reduced lipophilic VdW contacts and excessive flexibility, possibly explaining the high (about a hundred times), but not exceptional α4β2 vs. human-α3β4 selectivity of the pyridyl ether **Sazetidine**-**A** and of our most selective compounds. 

Based on these premises, a systematic docking analysis of the **A**-**84543** analogues was conducted into the cryo-EM structures of the two human subtypes drawing the following conclusions about the α4β2 affinity and the over α3β4 selectivity of the lead compounds reported in Figure 1.
**A**-**84543** and (*S*)-**nicotine** interact with the so-called aromatic box through the pyrrolidine N^+^ in both subtypes and with β2-Phe119 through the pyridine ring. In the α3β4 binding site, due to the absence of phenylalanine in the β4 side, the pyridine of the flexible **A**-**84543** “falls out” from the plane occupied in the α4β2 binding site whereas the nicotine pyridine maintains a substantially unchanged disposition. This is consistent with the high selectivity of **A**-**84543** and the only moderate one of nicotine.The docking of the 3-hydroxy phenyl ether (*S*)-**3** is very similar to that of (*S*)-**nicotine**: Overlap of the pyrrolidine rings, same interactions with the β2 hydrophilic sub-pocket through the *m*-OH rather than structural water and same minimal change of the disposition of the aromatic ring in the two subtypes. This is consistent with the only moderate selectivity displayed by both compounds.For the hydroxypyridyl ether (*S*)-**7**, α4β2 super-agonist with almost null α3β4 activity but with an affinity and selectivity profiles very similar to those of nicotine, a hydroxyphenyl ether-like extended pose without water-mediated interaction and an **A**-**83543**-like folded pose with pyridine nitrogen interacting with structural water can be assumed in both subtypes. Such a dualism could be correlated with the significantly different degree of selectivity, typical of this compound, in binding and functional tests.The 3,5-disubstituted phenyl ethers show the conventional α4β2 interactions (N^+^-aromatic box, phenyl- β2-Phe119) and, in the case of the 3-hydroxy-5-(6-hydroxyhexinyl)phenyl ether (*S*)-**15**, an additional interaction of the chain’s terminal OH with an α4 glutamic acid residue. Such interaction with the corresponding identical α3 aminoacidic residue is lost in the α3β4 binding site. This is consistent with the high selectivity (131 times) of the above derivative.The (*S*,*R*) diastereomer of the pyrrolidinyl pyridodioxane with N in place of benzodioxane C(5)H ((*S*,*R*)-**33**) accommodates its N^+^ in the aromatic box of the α4β2 binding site and establishes a pyridine-β2-Phe119 interaction and a water-mediated interaction with the hydrophilic β2 sub-pocket through the pyridine nitrogen. These two interactions are lost when the molecule is docked into the α3β4 binding site. Such binding modes are conserved and favoured in (*S*,*R*)-**14**, where pyridodioxane deconstruction to α-methyl prolinol pyridyl ether corresponds to a significant increase in α4β2 affinity (410 nM → 27 nM) and selectivity (40 → 185 times).The high α4β2 affinity of 7-OH and 7-NH_2_ substituted pyrrolidinyl benzodioxanes (*S*,*R*)-**2** and (*S*,*R*)-**29** is traceable to that of the prolinol 3-hydroxyphenyl ether (*S*)-**3**, without involvement of structural water. On the other hand, the significantly lower or null selectivity could be ascribed to a less detrimental effect of Phe to Leu replacement for the rigid and extended benzodioxane scaffold compared with flexible aryloxymethyl residues.With opposite stereo-preference concerning the dioxane stereocenter, the 5-amino substitution at benzodioxane ((*S*,*S*)-**37**) maintains high α4β2 affinity thanks to the same productive interactions established by the above mentioned 7-substituted pyrrolidinyl benzodioxanes, but, notably, it does not confer the ability to interact with the wider β4 minus side resulting in high selectivity.

In a nutshell, comparative docking into the two human α4β2 and α3β4 nAChR structures of this large series of pyrrolidine based α4β2 ligands indicates that constantly discriminating factors for the α4β2 vs. α3β4 subtype selectivity are the stabilization of the ligand’s aromatic ring by the not conserved β2-Phe119 residue and direct or water mediated interaction with hydrophilic residues of the β minus side conditioned by decoration of the aromatic ring and extensibility of the substructure of which it is part (Figure 12). The remarkable subtype selectivity achieved by some derivatives, such as the 5-amino substituted pyrrolidinyl benzodioxane (*S*,*S*)-**37**, the semirigid α-methyl prolinol pyridyl ether (*S*,*R*)-**14,** and the hydroxyhexinyl phenol derivative (*S*)-**15**, suggests that such discrimination can be highly effective. 

## Figures and Tables

**Figure 1 molecules-26-03603-f001:**
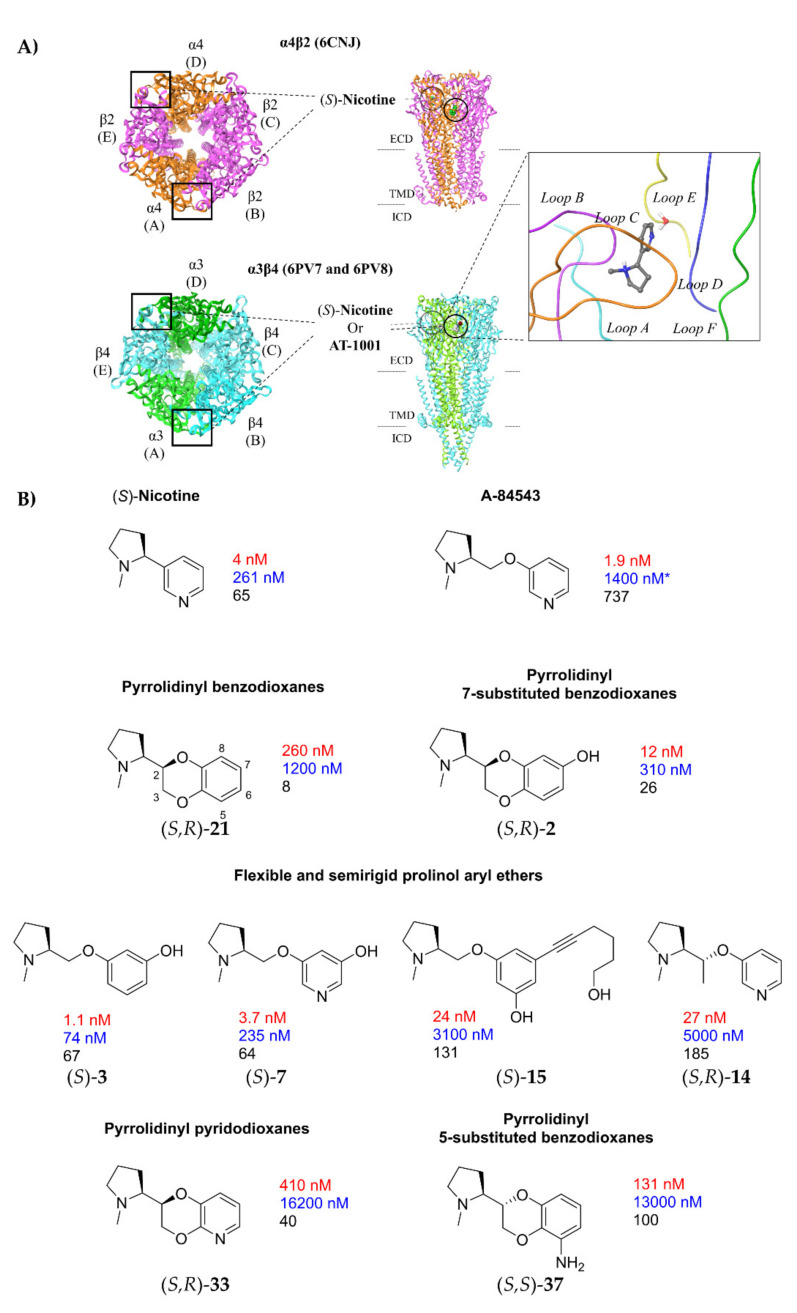
(**A**) Schematic overview of the α4β2 and α3β4 nAChRs architecture (adapted from 6CNJ, 6PV7, and 6PV8). The α-β localization of the orthosteric binding sites, where (*S*)-nicotine (6CNJ or 6PV7) and AT-1001 (6PV8) bind, are shown. Table 6. PV7, together with an overview of the loop composition of the binding site, is represented. (**B**) Binding affinities (K_i_, nM) for rat α4β2 nAChR (in red) and for human α3β4 nAChR (in bleu) and α4β2 vs. α3β4 binding selectivity (in black) of some title compounds exhibiting the highest α4β2 affinities. (*S*)-nicotine and A-84543 are reported for comparison. Asterisk on α3β4 affinity of A-84543 indicates that it was determined at the rat subtype.

**Figure 2 molecules-26-03603-f002:**
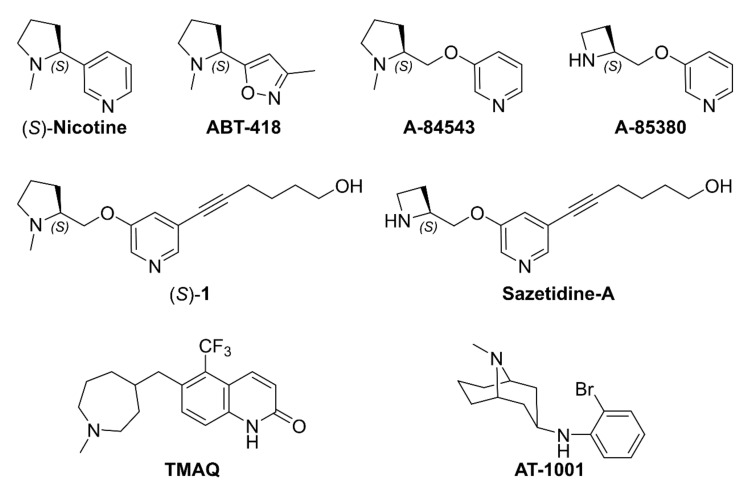
Representative structures of α4β2 and α3β4 ligands from the literature, providing useful information regarding inter-species differences of α4β2 vs. α3β4 selectivity.

**Figure 3 molecules-26-03603-f003:**
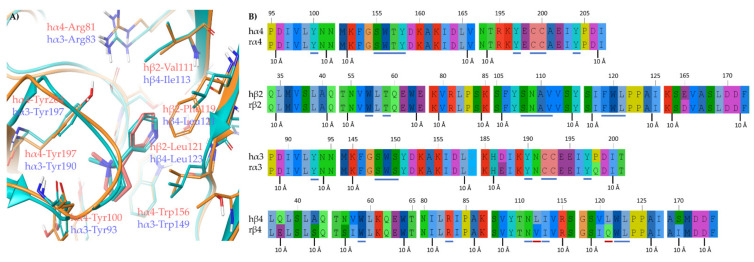
(**A**) Superimposition of the hα4β2 and hα3β4 binding sites complexed with (*S*)-**nicotine** (orange residues, backbone cartoons, and red ligand for hα4β2 and light blue residues, backbone cartoons, and ligand for hα3β4) extracted from the cryo-EMs 6CNJ and 6PV7, respectively. (**B**) Extracts of alignments of human and rat α4, β2, α3, and β4. Residues within 5 Å from the ligand has been underlined in blue, when identical between the species, in red when different.

**Figure 4 molecules-26-03603-f004:**
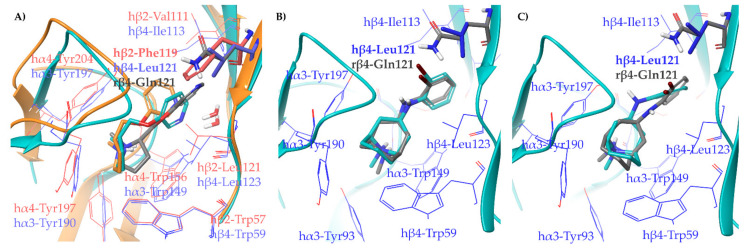
Receptor backbones are represented by orange (hα4β2) or cyan (α3β4) cartoons. (**A**) Comparison between the proposed binding modes of **A**-**84543** at the hα4β2 (PDB ID: 6CNJ, orange ligand and red residues), hα3β4 (PDB ID: 6PV7, cyan ligand and blue residues), and at the rα3β4 (grey ligand and residue). (**B**) Comparison between the original cryo-EM binding mode of **AT**-**1001** at the AB interface of hα3β4 (PDB ID: 6PV8, cyan ligand and blue residues) with the docked binding pose at the rα3β4 (grey ligand and residue) and (**C**) at the DE interface.

**Figure 5 molecules-26-03603-f005:**
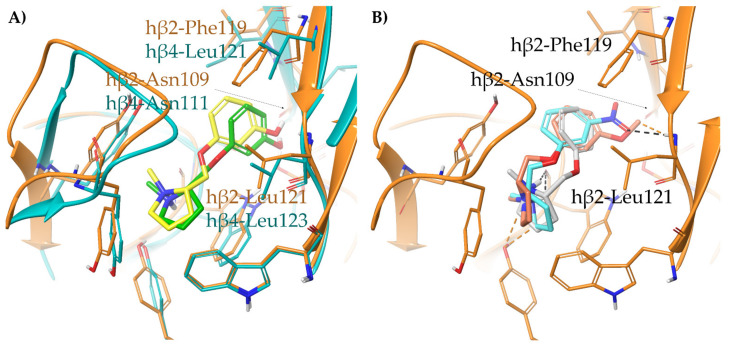
(**A**) Comparison of the proposed binding mode of the high affinity α4β2 ligand (*S*)-**3** at the hα4β2 (yellow ligand, orange residues and orange backbone cartoon, PDB ID: 6CNJ) and at the hα3β4 (green ligand, light blue residues, and light blue backbone cartoons, PDB ID: 6PV7). (**B**) Docking of the weak binders (*S*)-**4** (light grey), (*S*)-**5** (faded orange), and (*S*)-**6** (cyan) at the hα4β2 (orange residues and orange backbone cartoon, PDB ID: 6CNJ). Unfavorable steric clashes and hydrogen bonds are depicted as dashed orange and black lines, respectively.

**Figure 6 molecules-26-03603-f006:**
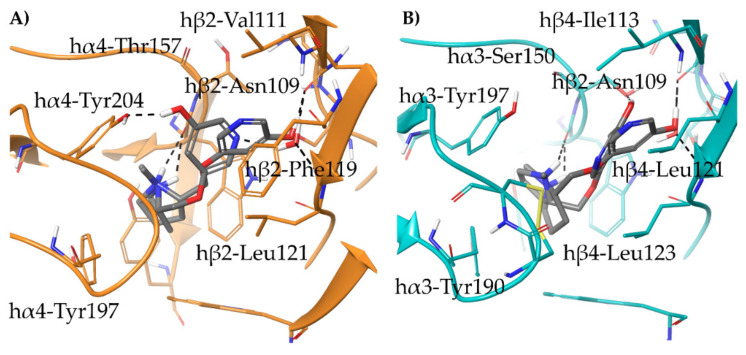
Proposed binding modes of (*S*)-**7** (dark grey) at the (**A**) hα4β2 (PDB ID: 6CNJ) and (**B**) at the hα3β4 (PDB ID: 6PV7) binding sites, both including and removing the structural water (kinked and extended conformation, respectively). Receptor backbones are represented by orange (hα4β2) and cyan (hα3β4) cartoons. Hydrogen bonds are depicted as black dashed lines.

**Figure 7 molecules-26-03603-f007:**
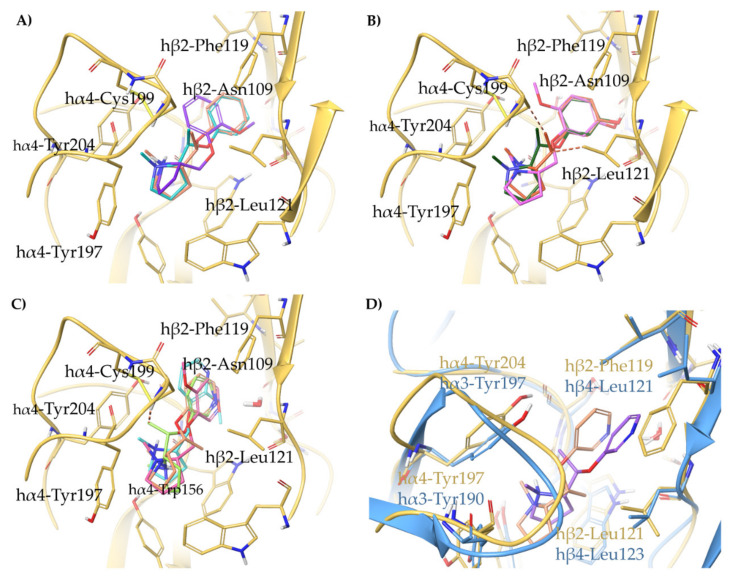
Receptor backbone and residues are depicted in yellow (hα4β2) and light blue (hα3β4). Unfavourable steric clashes are represented with orange dashed lines. Proposed binding modes at hα4β2 nAChR (adapted from 6CNJ) of the (**A**) semirigid non-hydroxylated phenyl ethers (*S*)-**8** (purple), (*S*,*R*)-**10** (cyan), and (*S*,*S*)-**10** (faded orange); (**B**) semirigid hydroxylated phenyl ethers (*S*)-**9** (pink), (*S*,*R*)-**11** (dark green), and (*S*,*S*)-**11** (orange); and of (**C**) semirigid piridyl ethers (*S*)-**12** (light cyan), (*S*)-**13** (faded pink), (*S*,*R*)-**14** (faded orange), and (*S*,*S*)-**14** (yellow-green). (**D**) Comparison of the binding modes of (*S*,*R*)-**14** at the hα4β2 (ligand faded orange) and at the hα3β4 (ligand faded purple) nAChRs.

**Figure 8 molecules-26-03603-f008:**
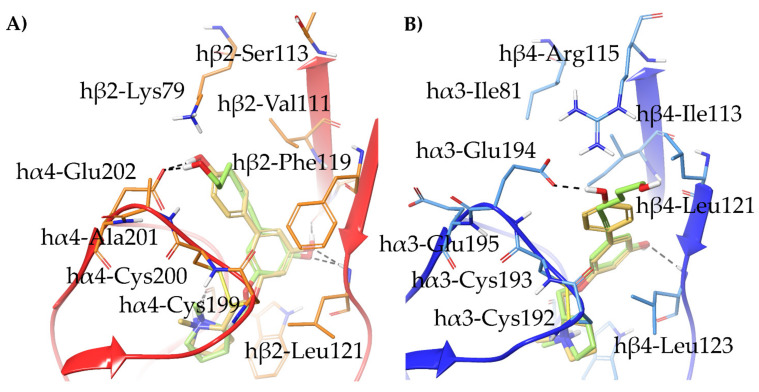
Proposed binding modes of the selective α4β2 ligand (*S*)-**15** (yellow-green) and of the unselective (*S*)-**16** (yellow) at the (**A**) hα4β2 nAChR and at the (**B**) hα3β4 nAChR. Receptor backbones are represented with red (hα4β2) and blue (hα3β4) cartoons, while hα4β2 and hα3β4 residues are shown in orange and light blue, respectively. Hydrogen bonds are depicted with black dashed lines.

**Figure 9 molecules-26-03603-f009:**
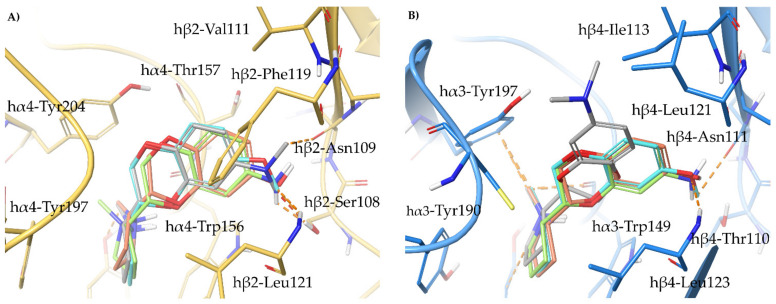
Binding poses of the high affinity ligands (*S*,*R*)-**2** (faded orange), (*S*,*R*)-**29** (yellow-green), and of the weak binders (*S*,*R*)-**26** (cyan), (*S*,*R*)*-***31** (grey) at the (**A**) hα4β2 nAChR (adapted from 6CNJ, yellow residues and backbone cartoons) and at the (**B**) hα3β4 nAChR (adapted from 6PV7, light blue residues and backbone cartoons). Unfavorable and very unfavorable steric clashes are depicted with orange and red dashed lines, respectively.

**Figure 10 molecules-26-03603-f010:**
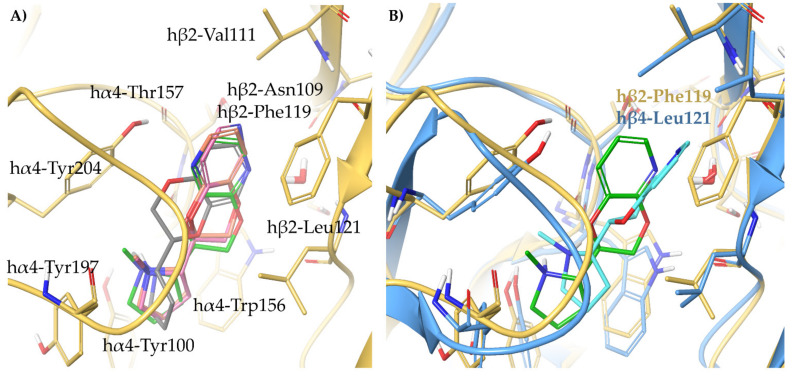
(**A**) Binding modes of the pyridodioxane-based selective hα4β2 partial agonist (*S*,*R*)-**33** (green) and of the weak binders (*S*,*R*)-**34** (pink), (*S*,*R*)-**35** (grey), and (*S*,*R*)-**36** (faded orange) at the hα4β2 binding site (adapted from 6CNJ, yellow residues, and yellow backbone cartoons). (**B**) Superimposition of (*S*,*R*)-**33** bound at the hα4β2 binding site (green ligand, yellow residues, and yellow backbone cartoons, adapted from 6CNJ) and bound at hα3β4 binding site (cyan ligand, light blue residues, and backbone cartoons, adapted from 6PV7).

**Figure 11 molecules-26-03603-f011:**
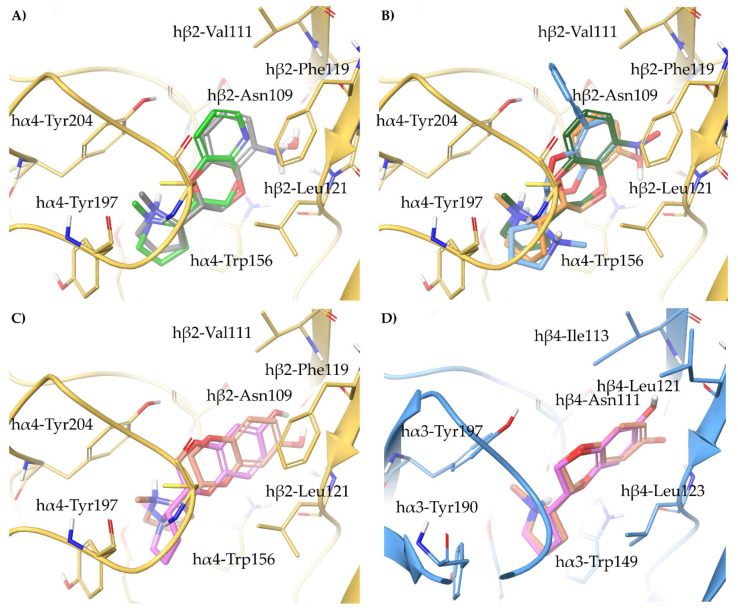
(**A**) Similar binding poses of (*S*,*R*)-**33** (green) and (*S*,*S*)-**37** (grey) at the hα4β2 nAChR, retraced by (**B**) (*S*,*S*)-**38** (dark green) and (*S*,*S*)-**39** (orange), but not by (*S*,*S*)-**40** (light blue). (**C**) Superimposition of the binding modes of (*S*,*R*)-**2** (faded orange) and (*S*,*R*)-**43** (pink) at the hα4β2 and (**D**) hα3β4 nAChRs binding sites.

**Figure 12 molecules-26-03603-f012:**
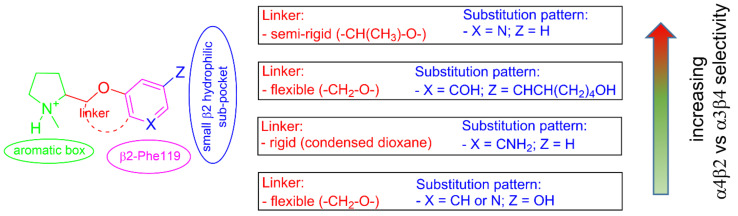
Crucial observations regarding α4β2 vs. α3β4 selectivity.

**Table 1 molecules-26-03603-t001:** Binding affinity data at α4β2 and α3β4 nAChRs and α4β2 vs. α3β4 selectivity ratio of compounds (*S*,*R*)-**2**, (*S*)-**3**–**7** reported in the literature. ^a^ Tested at rat cortex using [^3^H]-epibatidine, unless otherwise specified. ^b^ Tested at membranes of human α3β4 transfected cells. ^c^ Data from Bavo et al. [33]. ^d^ Data from Bolchi et al. [32]. ^e^ Tested at whole at brain preparation using [^3^H]-cytisine; from Elliot et al. [44]. ^f^ Data from Bolchi et al. [7].

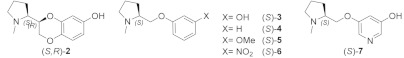
Compound	α4β2 K_i_ (μM) ^a^	α3β4 K_i_ (μM) ^b^	α3β4 K_i_/α4β2 K_i_
(*S*,*R*)-**2** ^c^	0.012	0.310	25.8
(*S*)-**3** ^d^	0.0011	0.074	67.3
(*S*)-**4** ^e^	0.042	n.a.	n.a.
(*S*)-**5** ^f^	0.600	4.5	7.5
(*S*)-**6** ^f^	0.0312	0.946	30.3
(*S*)-**7** ^d^	0.0037	0.235	63.5

**Table 2 molecules-26-03603-t002:** Binding affinity data at α4β2 and α3β4 nAChRs and α4β2 vs. α3β4 selectivity ratio of compounds **8**–**14** reported in Bolchi et al. [32]. ^a^ Tested at rat cortex using [^3^H]-epibatidine. ^b^ Tested at membranes of human α3β4 transfected cells.

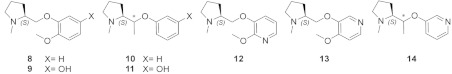
Compound	α4β2 K_i_ (μM) ^a^	α3β4 K_i_ (μM) ^b^	α3β4 K_i_/α4β2 K_i_
(*S*)-**8**	9.4	0.749	0.08
(*S*)-**9**	0.0189	0.271	14.3
(*S*,*R*)-**10**	1.55	1.3	0.84
(*S*,*R*)-**10** + (*S*,*S*)-**10**	4.59	1.4	0.31
(*S*,*R*)-**11**	0.011	0.257	23.4
(*S*,*S*)-**11**	0.192	0.752	3.9
(*S*)-**12**	7.28	0.794	0.11
(*S*)-**13**	0.255	2.1	8.2
(*S*,*R*)-**14**	0.027	5.0	185
(*S*,*S*)-**14**	0.877	5.6	6.4

**Table 3 molecules-26-03603-t003:** Binding affinity data at α4β2 and α3β4 nAChRs and α4β2 vs. α3β4 selectivity ratio of compounds **15**–**20** reported by Bolchi et al. [7]. ^a^ Tested at rat cortex using [^3^H]-epibatidine. ^b^ Tested at membranes of human α3β4 transfected cells.

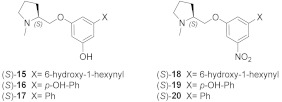			
Compound	α4β2 K_i_ (μM) ^a^	α3β4 K_i_ (μM) ^b^	α3β4 K_i_/α4β2 K_i_
(*S*)-**15**	0.0237	3.1	130.8
(*S*)-**16**	0.0038	0.030	7.9
(*S*)-**17**	0.528	0.200	0.4
(*S*)-**18**	0.0142	1.200	84.5
(*S*)-**19**	0.012	0.122	10.2
(*S*)-**20**	0.330	0.947	2.9

**Table 4 molecules-26-03603-t004:** Binding affinity data at α4β2 and α3β4 nAChRs and α4β2 vs. α3β4 selectivity ratio of compounds **2**, **21**–**32**. ^a^ Tested at rat cortex using [^3^H]-epibatidine. ^b^ Tested at membranes of human α3β4 transfected cells. ^c^ Data from Pallavicini et al. [24]. ^d^ Data from Bolchi et al. [5]. ^e^ Data from Bolchi et al. [31]. ^f^ Data from Bavo et al. [33].

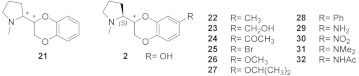			
Compound	α4β2 K_i_ (μM) ^a^	α3β4 K_i_ (μM) ^b^	α3β4 K_i_/α4β2 K_i_
(*R*,*R*)-**21**	43.8 ^c^	-	-
(*R*,*S*)-**21**	12.5 ^c^	-	-
(*S*,*R*)-**21**	0.26 ^c^	1.2 ^d^	4.6
(*S*,*S*)-**21**	0.47 ^c^	8.2 ^d^	17.4
(*S*,*R*)-**2** ^e^	0.012	0.310	25.8
(*S*,*S*)-**2** ^e^	0.421	0.7	1.7
(*S*,*R*)-**22** ^e^	42	-	-
(*S*,*R*)-**23** ^e^	51	-	-
(*S*,*R*)-**24** ^e^	97	-	-
(*S*,*R*)-**25** ^e^	8.1	-	-
(*S*,*R*)-**26** ^e^	17	-	-
(*S*,*R*)-**27** ^e^	9.3	-	-
(*S*,*R*)-**28** ^e^	35	-	-
(*S*,*R*)-**29** ^f^	0.022	0.019	0.9
(*S*,*R*)-**30** ^f^	14	4.5	0.3
(*S*,*R*)-**31** ^f^	147	2.2	0.02
(*S*,*R*)-**32** ^f^	6.5	1.9	0.3

**Table 5 molecules-26-03603-t005:** Binding affinity data at α4β2 and α3β4 nAChRs and α4β2 vs. α3β4 selectivity ratio of compounds **33**–**36**, from Bolchi et al. [5]. ^a^ Tested at rat cortex using [^3^H]-epibatidine. ^b^ Tested at membranes of human α3β4 transfected cells.

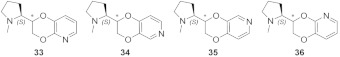
Compound	α4β2 K_i_ (μM) ^a^	α3β4 K_i_ (μM) ^b^	α3β4 K_i_/α4β2 K_i_
(*S*,*R*)-**33**	0.41	16.2	39.5
(*S*,*S*)-**33**	30.4	22	0.7
(*S*,*R*)-**34/**(*S*,*S*)-**34**	2.5	12.3	4.9
(*S*,*R*)-**35**	1.64	5.8	3.5
(*S*,*S*)-**35**	3.6	8.9	2.5
(*S*,*R*)-**36**	43	-	-
(*S*,*S*)-**36**	30	-	-

**Table 6 molecules-26-03603-t006:** Binding affinity data at α4β2 and α3β4 nAChRs and α4β2 vs. α3β4 selectivity ratio of compounds **37**–**44**, from Bavo et al. [33]. ^a^ Tested at rat cortex using [^3^H]-epibatidine. ^b^ Tested at membranes of human α3β4 transfected cells.

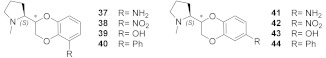
Compound	α4β2 K_i_ (μM) ^a^	α3β4 K_i_ (μM) ^b^	α3β4 K_i_/α4β2 K_i_
(*S*,*R*)-**37**	7.1	3.9	0.5
(*S*,*S*)-**37**	0.131	13	100
(*S*,*R*)-**38**	12.2	1.1	0.1
(*S*,*S*)-**38**	0.335	6.7	20.0
(*S*,*R*)-**39**	0.55	3.9	7.1
(*S*,*S*)-**39**	1.7	>100	>59
(*S*,*R*)-**40**	0.64	2.7	4.2
(*S*,*S*)-**40**	7.3	5.3	0.7
(*S*,*R*)-**41**	86	3.5	0.04
(*S*,*R*)-**42**	46	15	0.3
(*S*,*R*)-**43**	50	1.9	0.04
(*S*,*R*)-**44**	59	5.4	0.1

## Data Availability

Data presented in this study are available on request from the corresponding author.

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
