# Peer review of "Determinants for α4β2 vs. α3β4 Subtype Selectivity of Pyrrolidine-Based nAChRs Ligands: A Computational Perspective with Focus on Recent cryo-EM Receptor Structures"

_molecules, 2021, doi:10.3390/molecules26123603_

Round 1

Reviewer 1 Report

The authors docked pyrrolidine-based ligands to the recently reported cryo-EM structure of nicotinic acetylcholine receptors (nAChRs) using a computational method and described their potential interactions and selectivity. This study explained the selectivity for the docking based on the biochemical analysis results of previously known ligands, and this result is thought to provide useful information for the future development of nAchRs substrates and new drugs. Meanwhile, in previous cryo-EM structural studies, the positions of some atoms were ambiguous in (i) Density Map for Nicotine and Water and (ii) Density Map for AT-1001 at Site 1 and Site 2. This may be the reason the cryo-EM resolution is low, or the high B-factor may be high. The docking results obtained from this manuscript are considered to be extracts for theoretically stabilized ones because computational techniques are used, but to prove this, more robust experimental results (e.g. mutagenesis based on the results based on docking) need to be supported the authors’ clams.

In order to improve the manuscript, I suggest the following should be modified.

  1. The author should write more details about the docking method.
  2. There are many duplicates in the method (eg. were docked in the rα3β4 as described in paragraph 3.2). It is necessary to combine subsections to improve readability.
  3. The resolution of the figure is low, and the residue notation is small. Making the cartoon transparent will make it clearer that the molecule the author wants to show. It also suggests removing unnecessary main chain.

Reviewer 2 Report

Dear Author

The article entitled "Determinants for α4β2 vs α3β4 subtype selectivity of pyrrolidine-based nAChRs ligands: a computational perspective with focus on recent cryo-EM receptor structures" is interesting and provides new knowledge in the area of ​​interest.
I have had the pleasure of reading the article, from the point of view of my experience in molecular modeling.
The authors used molecular docking techniques to support their conclusions, relating them to the experimental results.
The authors were very detailed in the elaboration of the molecular docking model as well as in the analysis of the results. They considered the effect of water, when comparing systems with and without solvent. Despite this technique being limited in the prediction of the binding mode, I consider that the model and the analyzes are sufficient to support its conclusions.
Otherwise the work is well written and provides new knowledge in the area. For this reason I recommend it for publication.

Reviewer 3 Report

The authors describe structural determinants important for selectivity between nicotine receptor a4b2 and a3b4 subtypes using in-silico docking studies. To delineate the determinants they make use of forty known a4b2 pyrrolidine ligands and docked them into newly solved cryo-electron 3D protein structures of the relevant nicotine receptors subtypes. They identified the b2-phenilalanine-119 and the direct or water mediated interaction with hydrophilic residues of the b-minus receptor part as key distinctive feature for the selectivity of compounds. Moreover, they provide a rigorous comparison of the human versus rat nicotine receptor differences. I believe this is quality work, which was done very systematically, and will offer medicinal chemists important insight for designing novel selective a4b2 nAChR agonists. I do have some suggestions that I believe would make the work even stronger:

  • My main concern is the lack of a redocking validation. The authors should redock native ligands in the used PDB structures binding sites, using the same procedure as docking is done on tested ligands, and report the corresponding RMSDs. This is important for the validation of the docking method (see for example: https://www.mdpi.com/1999-4923/13/3/315/htm or https://pubs.acs.org/doi/abs/10.1021/acs.jcim.8b00981).

  • In the introduction I would recommend writing a sentence or two about the problems with non-selective compounds. What kind of side effects are problematic when a3b4 is also activated?

  • In the text you refer to specific numbering of the atoms in the compounds, e.g. on page 2, line 50. It would be beneficial for the clarity of the work if you could show this numbering in Figure 1 for representative structures.

  • When introducing compounds, you should always refer to the appropriate figure. For example when introducing (S)-1, you do not refer to Figure 2.

  • Could you please explicitly explain in the work, why you compare to rats Ki values for a4b2 instead of humans? I imagine no measurements yet exists for the human subtype? However, this should be specifically stated. Also on page 4, line 121, you do refer to binding affinities to the human a4b2 subtype to sazetidine-A.

  • You state that the ha3b4 binding site is less compact due to a 2.1 A outward displacement of loop C. It would be good if you could label explicitly this loop in a Figure. Moreover, I would suggest that in the introduction you present in a general fashion the two PDB 3D structures used in the study. It would be beneficial to the reader to, for example, present the location of the binding site(s) on a Figure presenting the whole structure, label the loops, the later mentioned AB, DE interfaces, show the interactions between the structural water and amino-acids, present the parts of the binding site. e.g. the aromatic box, etc…

  • It would be good to calculate the RMSD values for all cases when the same ligand is docked and compared, like the cases in Figure 4.

  • The case presented in Figure 4B and 4C is interesting. You report different binding poses, between the AB and DE interfaces. However, the interface look extremely similar. Could it be, that the difference is just the consequence of stochastic nature of the docking algorithm. In connection to this, I would recommend you repeat the docking experiment with different random seeds, and see if you still get this difference.

  • In Figures you should consistently show H-bonding and pi-pi interactions. Moreover, you should always present amino acid residues in Figures that are mentioned in the corresponding text.

  • On page 7, Line 239, you talk about b2-Asn119. Shouldn’t this residue be Phe119?

  • In the conclusion, page 17, line 577, you state that rat b4-Gln121 disfavors advantageous interactions even more that hB4-Leu121. Could you explain, why do you believe this is the case (steric clashing, unfavorable same-charge repulsion, …?)?.

  • Due to quite a large amount of results presented, it would be beneficial if the authors could add a table after each 2.# subsection, which would contain the overview of the interactions of each docked compound, stating the numbered ligand atom(s), amino-acid residue, and type of interaction. Or alternatively, add one larger table at the end.

  • A Figure overviewing crucial observations in connection to activity and selectivity would be beneficial, maybe something similar to Figure 4 in https://pubs.acs.org/doi/full/10.1021/acs.jmedchem.9b01886.

Round 2

Reviewer 1 Report

The author has addressed my concerns, and through the revision, the revised manuscript has been mproved compared to the original manuscript, and I support its publication.

Reviewer 3 Report

I believe the authors were very diligent and rigorous in their review and I would recommend publication.

I would only recommend to round the RMSD values to the first decimal.